# Cardiovascular Damage in COVID-19: Therapeutic Approaches Targeting the Renin-Angiotensin-Aldosterone System

**DOI:** 10.3390/ijms21186471

**Published:** 2020-09-04

**Authors:** Jairo Lumpuy-Castillo, Ana Lorenzo-Almorós, Ana María Pello-Lázaro, Carlos Sánchez-Ferrer, Jesús Egido, José Tuñón, Concepción Peiró, Óscar Lorenzo

**Affiliations:** 1Laboratory of Diabetes and Vascular pathology. Instituto de Investigaciones Sanitarias-Hospital Fundación Jiménez Díaz. Universidad Autónoma, 28040 Madrid, Spain; jairo.lumpuy@estudiante.uam.es (J.L.-C.); jegido@fjd.es (J.E.); jtunon@fjd.es (J.T.); 2Department of Internal Medicine. Hospital Fundación Jiménez Díaz, 28040 Madrid, Spain; alorenzoa@fjd.es; 3Department of Cardiology. Hospital Fundación Jiménez Díaz, 28040 Madrid, Spain; ampello@quironsalud.es; 4Department of Pharmacology, School of Medicine, Universidad Autónoma de Madrid, 28049 Madrid, Spain; carlosf.sanchezferrer@uam.es (C.S.-F.); concha.peiro@uam.es (C.P.); 5Spanish Biomedical Research Centre on Diabetes and Associated Metabolic Disorders (CIBERDEM) Network, 28029 Madrid, Spain

**Keywords:** COVID-19, cardiovascular, RAAS, ACE2/Ang-(1–7)/MasR

## Abstract

Coronavirus disease 2019 (COVID-19) is usually more severe and associated with worst outcomes in individuals with pre-existing cardiovascular pathologies, including hypertension or atherothrombosis. Severe acute respiratory syndrome coronavirus-2 (SARS-CoV-2) can differentially infect multiple tissues (i.e., lung, vessel, heart, liver) in different stages of disease, and in an age- and sex-dependent manner. In particular, cardiovascular (CV) cells (e.g., endothelial cells, cardiomyocytes) could be directly infected and indirectly disturbed by systemic alterations, leading to hyperinflammatory, apoptotic, thrombotic, and vasoconstrictive responses. Until now, hundreds of clinical trials are testing antivirals and immunomodulators to decrease SARS-CoV-2 infection or related systemic anomalies. However, new therapies targeting the CV system might reduce the severity and lethality of disease. In this line, activation of the non-canonical pathway of the renin-angiotensin-aldosterone system (RAAS) could improve CV homeostasis under COVID-19. In particular, treatments with angiotensin-converting enzyme inhibitors (ACEi) and angiotensin-receptor blockers (ARB) may help to reduce hyperinflammation and viral propagation, while infusion of soluble ACE2 may trap plasma viral particles and increase cardioprotective Ang-(1–9) and Ang-(1–7) peptides. The association of specific ACE2 polymorphisms with increased susceptibility of infection and related CV pathologies suggests potential genetic therapies. Moreover, specific agonists of Ang-(1–7) receptor could counter-regulate the hypertensive, hyperinflammatory, and hypercoagulable responses. Interestingly, sex hormones could also regulate all these RAAS components. Therefore, while waiting for an efficient vaccine, we suggest further investigations on the non-canonical RAAS pathway to reduce cardiovascular damage and mortality in COVID-19 patients.

## 1. Introduction

Coronavirus disease 2019 (COVID-19) is a global pandemic caused by the new severe acute respiratory syndrome coronavirus-2 (SARS-CoV-2). Compared to the previous SARS-CoV-1 and the Middle East Respiratory Syndrome coronavirus (MERS) originated in 2003 and 2012 respectively [1,2], SARS-CoV-2 is less lethal, though it has claimed more lives than both combined due to the higher transmission rate [3]. However, the epidemiology of SARS-CoV-2 remains largely unknown. Most publications are small, single-center studies, and focused on hospitalized patients. Consequently, the ability to accurately estimate infectivity, symptom burden, and non-fatal and fatal complications is limited, leading to a lack of an effective therapy. Also, a substantial undocumented infection (86% of all cases) that facilitates the rapid dissemination of the virus has been documented [4].

SARS-CoV-2 infection initiates in the respiratory system, when the S protein of its external layer binds the angiotensin-converting enzyme-2 (ACE2) at the plasma membrane of host cells [5]. This interaction also needs the cellular serine protease transmembrane serine protease-2 (TMPRSS2) to prime the S protein, allowing the fusion of membranes [6]. SARS-CoV-2 might also bind to other potential receptors such as dipeptidyl peptidase-4, sialoglycans, or the extracellular matrix metalloproteinase inducer (EMMPRIN-CD147), enhancing viral infectivity [7,8]. Moreover, after cleavage of the short for a disintegrin and metalloproteinase domain 17 (ADAM-17), ACE2 can be released to the circulation, keeping the ability to interact with viral particles (see below) [9]. Importantly, ACE2 has been mainly localized in lung alveolar epithelium and gastrointestinal tract, but also in bone marrow, kidneys, spleen, myocardium, and the vasculature [10]. Its ubiquitous location explains the multi-organ injury affectation, especially dangerous in the cardiovascular (CV) system [11]. Also, high levels of plasma leukocytes and an enhanced neutrophil/lymphocyte ratio have been described in post-infection stages [12]. Both cellular and humoral immune mechanisms are responsible for this hyperinflammatory response. T-cells initiate a cytokine storm by releasing proinflammatory cytokines (i.e., TNF-α, interleukins IL-1β, IL-18, and IL-6, interferon gamma IFN-γ, and MCP-1), and reducing anti-inflammatory factors (i.e., IL-10, IFN-γ-induced protein-10; IP-10, IL receptors; IL-2Rα, IL-1Ra) [13]. However, SARS-CoV-2 can also infect and kill T-cells and peripheral white blood cells, leading to immune deficiency [14]. In addition, viral RNA activates the innate immune system through two main major classes of pattern recognition receptors. The toll-like receptors (TLRs)-7 and -8 trigger downstream signaling effectors like nuclear factor-κB (NF-κB) [15], which enhances the expression of cytokines, which activate T-cells for secretion of IFN-γ and IL-17. Also, Nucleotide-binding and oligomerization domain (NOD)-like receptors induce IL1β upregulation via the NOD-, LRR- and pyrin domain-containing protein-3 (NLRP3)-inflammasome [16]. In turn, this complex cleaves pro-caspase-1 into caspase-1 to activate cell death by pyroptosis [17]. Consequently, the uncontrolled release of pro-inflammatory factors and the unbalanced immune response produce an aggressive inflammation and injury in peripheral tissue [18].

### 1.1. SARS-CoV-2 and the CV System

Around 80% of COVID-19 patients exhibit a mild form of the disease, whereas 5% of them are presenting a severe pathology with respiratory failure, septic shock, and multi-organ dysfunction, with half of these being at extreme risk of death [19]. The primary causes of morbidity and mortality in COVID-19 are lung injury with acute respiratory stress syndrome (ARDS), thromboembolism, and heart failure or cardiac dysfunction [20,21]. Also, the most prevalent comorbidities are hypertension (15.6%), diabetes (7.7%), and CV diseases (4.7%) [22]. The heterogeneity of responses among individuals with SARS-CoV-2 infection may depend on the development of CV pathologies [23]. In most cases, CV abnormalities occur around two weeks after infection, suggesting that they may not be caused by a direct viral infection but rather by a hyperinflammatory reaction from hosts cells [24]. However, due to the systemic nature of the disease and the lack of CV studies, especially histopathological examinations, the diagnosis of CV damage induced by SARS-CoV-2 is insufficient. Endothelial and perivascular cells, and cardiomyocytes and cardiac macrophages, can be dramatically altered by SARS-CoV-2, leading to irreversible consequences [25]. Both endothelial dysfunction and acute myocardial injury are central events in the CV failure of COVID-19 patients [26].

#### 1.1.1. Endothelial Dysfunction

Endothelial dysfunction is characterized by a change in the physiological actions of the endothelium towards a vasoconstrictor, proinflammatory, and prothrombic state, which frequently precede the development of CV disease [27]. COVID-19 has been considered as an eminently endothelial disease, though the effects of SARS-CoV-2 on vascular homeostasis are still poorly understood (Figure 1). In post-mortem samples from COVID-19 patients, a diffuse endothelial inflammation (endothelitis) with accumulation of inflammatory cells has been observed [28]. After infection, endothelial cells become inflamed and activated by the excessive release of cytokines derived from the exaggerated immune response [29]. Then, endothelial NF-κB is activated and propagates inflammation by upregulation of a wide range of pro-inflammatory cytokines, chemokines, and adhesion molecules, promoting recruitment of leukocytes. Persistent inflammation causes endothelial dysfunction and profound vascular disturbances [29]. Moreover, the alteration of cell adherent junctions produces vascular permeability and edema. Of interest, some evidence suggests that endothelial cell activation might also be directly induced by SARS-CoV2 particles. The endothelium expresses high levels of ACE2, TMPRSS2, and CD147 [11,30,31], which provide a dock-way in cells for viral infection [32]. Also, perivascular cells that communicate with endothelial cells to upregulate inflammatory responses, such as pericytes, show high ACE2 levels and can potentially be infected, contributing to increased inflammation [25]. Indeed, post-mortem analysis revealed the presence of viral structures in vascular beds from different organs [28]. Thus, endothelial cells may propagate the inflammatory wave to the surrounding tissues, including lung and heart [33,34].

In addition, post-mortem analysis showed an increase of endothelial cell death [28]. In COVID-19, endothelial dysfunction may lead to cellular apoptosis and other types of cell death, such as pyroptosis and necrosis [28,35]. The loss of cells disrupts the endothelial barrier and increases vascular leakage and protein extravasation. It also allows for the exposure of sub-endothelial collagen to circulating platelets, which initiate a pro-coagulant response [36]. Indeed, post-mortem specimens exhibited microvascular platelet-rich thrombotic depositions [37]. Venous thromboembolism in lungs is a common complication in severe COVID-19 [38], and arterial thromboembolism with ischemic events in large vessels or limb have been also reported, even in young patients [39,40]. Hypercoagulation and thrombosis are devastating consequences of SARS-CoV2 infection (Figure 1). Above 70% of non-survivor patients had findings compatible with disseminated intravascular coagulation (DIC) [41]. After injury or excessive activation, endothelial cells release pro-thrombotic and pro-coagulant molecules, including von Willebrand factor and tissue factor, while reducing anticoagulant agents such as antithrombin-III, protein C, or plasminogen activator inhibitor type-1 (PAI-1) [36]. Over-expression of platelet adhesion molecules (i.e., PECAM-1) and secretion of reactive oxidant species (ROS) also contribute to the formation of fibrin clots. In fact, a high proportion of hospitalized patients presented elevated concentrations of D-dimer, a degradation product of cross-linked fibrin that correlates with the severity of the disease and mortality [42]. Altogether, SARS-CoV-2 may be able to induce damage of endothelium and thereby initiate platelet aggregation and consecutive vessel occlusion. Finally, endothelial dysfunction shifts the vascular tone towards vasoconstriction. Pro-inflammatory mediators and vasoconstrictors (e.g., angiotensin-II (Ang-II)) stimulate intracellular signals that increase ROS formation, which quench nitric oxide (NO) and coenzyme tetrahydrobiopterin (BH4) as main vasodilator factors. The sustained vasoconstriction may lead to hypoperfusion, particularly relevant for pulmonary and cardiac function [43].

Hyperinflammation, DIC, and vasoconstriction contribute to the massive capillary congestion, collapse, and ischemia, as observed in lung, mesentery, and heart from COVID-19 individuals [34,44]. We believe that therapeutic stabilization of the endothelium may be essential during COVID-19, particularly in those patients with previous endothelial dysfunction due to comorbidities like hypertension, diabetes, and obesity [45,46].

#### 1.1.2. Acute Cardiac Injury

Acute cardiac injury has been reported to affect between 7% and 28% of hospitalized COVID-19 patients, being associated with worse outcomes and mortality, even in the absence of ARDS [47]. The diagnosis of cardiac injury was mainly approached by detection of specific plasma biomarkers. As early as 15 days after infection, cardiac troponin-I (cTnI) is elevated in most patients, with or without the presence of alterations in electrocardiogram and echocardiography [24,48]. Cardiac damage may be secondary to lung or systemic pathologies under an inflammatory milieu [23] (Figure 1). Vasculitis, endothelial dysfunction, or micro-thrombosis can damage the myocardium. As a reflection of the cytokine storm, patients showed a positive correlation between cTnI and inflammatory plasma biomarkers, such as D-dimer, ferritin, IL-6, and lactate dehydrogenase [49]. Particularly, IL-6 was also associated with the development of cardiac fibrosis and heart failure [50]. Furthermore, these patients displayed higher concentrations of plasma creatine kinase (CK)-MB, myohemoglobin, and N-terminal pro-B-type natriuretic peptide (NT-proBNP), in parallel to augmented leukocyte count, C-reactive protein, and procalcitonin [51]. Patients showed an atherosclerosis-like immune response with lower lymphocyte count but hyper-activated T-cells, mainly HLA-DR^+^, CD38^+^CD8^+^/CD4^+^, and CCR6^+^ Th17CD4^+^ [51]. In this sense, though there is no evidence of acute coronary syndrome or epicardial plaque rupture, these pathologies can be developed in COVID-19 subjects with or without previous atherosclerosis [52]. In fact, in a cohort of 18 hospitalized patients with ST-segment elevation on electrocardiography, 50% of them needed coronary angiography [53].

In addition, the presence of ACE2 receptors on the myocardium provides a theoretical mechanism for direct SARS-CoV-2 infection [23,54] (Figure 1). In SARS-CoV-1, viral RNA and monocyte/lymphocyte infiltration with stromal cardiac edema were detected at the myocardium in parallel to clinical myocarditis [55]. Thus, SARS-CoV-2 could also enter into cardiac cells, leading to a hyperinflammatory response within the cardiac tissue, together with myocarditis [56]. A recruitment of inflammatory cells typical of viral infection (CCR6+/Th17– and CD4+T–cells) has been observed in hearts from COVID-19 individuals [57]. However, more clinical evidence, such as the presence of viral particles inside cells or cardiomyocyte necrosis, has not been reported [58].

Furthermore, SARS-CoV-2 could infect resident macrophages that interact with conducting cells of the atrio-ventricular node expressing ACE2 receptors [59]. This contributes to viral spread and hyperinflammation and may alter cardiac rhythm. In COVID-19, patients with arrhythmias frequently suffered from myocardial injury and ischemia, hypoxia, widespread systemic inflammation, shock, electrolyte disturbances, and/or receive cardiotoxic drugs (i.e., hydroxychloroquine and antivirals). Arrhythmias were developed as a consequence of systemic illness and not only due to direct effects of infection [60]. Nevertheless, the prevalence of arrhythmias varies from different reports. In a study of 138 hospitalized patients, arrhythmias were exhibited in 17% of cases, whereas this percentage rose up to 44% when considering only patients admitted to an intensive care unit [61]. The most common electrocardiographic finding in COVID-19 patients was sinus tachycardia, whereas atrial fibrillation and monomorphic or polymorphic ventricular tachycardia were the most deleterious arrhythmias [62]. Atrial arrhythmias were, however, more frequent among patients requiring mechanical ventilation [63]. Finally, acute cardiac injury may be caused by stress cardiomyopathy as a result of hypoxia, increased metabolic demands, fever, or hypotension in the presence of severe infection and hyperinflammation [64]. The presence of type-2 myocardial infarction secondary to a mismatch between oxygen supply and metabolic demands was described in severe infection with sepsis or ARDS associated with hypoxia [65,66].

Altogether, the presence of myocarditis, arrhythmias, stress cardiomyopathy, myocardial infarction, inflammation, or vasculitis and endothelial dysfunction may favor the generation of myocardial dysfunction and, in turn, heart failure [23] (Figure 1). Remarkably, heart failure has been described in 23% of survivors and in 52% of non-survivors in a population of 191 COVID-19 patients [24].

### 1.2. Renin-Angiotensin-Aldosterone System (RAAS) Modulators in CV Damage of COVID-19

The RAAS is a complex system that plays an important role in the control of CV functions. Its overstimulation can increase vessel contraction and activate pro-hypertrophic and pro-fibrotic responses of CV cells [67]. The RAAS plays a crucial role in immune functions by regulating pro- and anti-inflammatory molecules, participating in cell recruitment. In the canonical pathway, the octapeptide hormone Ang-II is produced by a two-step cleavage process from angiotensinogen through renin and angiotensin convertase enzyme (ACE). To a lesser extent, Ang-II can also be converted by aminopeptidase-A to Ang-III, which conserves similar properties to the octapeptide [68]. Ang-II (and Ang-III) actions are mainly mediated by two distinct G protein-coupled angiotensin receptors, AT1R and AT2R. In the CV system, AT1R overactivity has been associated with the development of several pathological conditions, including hypertension, vascular inflammation, and remodeling, atherosclerosis, and heart failure [69]. Stimulation of AT1R triggers NF-κB to upregulate proinflammatory and pro-coagulant factors by resident cells and endothelial cells. Moreover, this ACE/Ang-II/AT1R axis also induces epithelial to mesenchymal transformation, which enhances pro-inflammatory factors and decreases endothelial permeability [69].

In the non-canonical axis of the RAAS, an ACE2-dependent deletion of Ang-II produces another RAAS peptide, termed Ang-(1–7). This heptapeptide can also be synthesized from Ang-I by neprilysin, or by degrading Ang-I into Ang-(1–9) and then to Ang-(1–7) by sequential actions of ACE2 and ACE [70]. Importantly, this secondary axis can counter-regulate many actions provoked by the ACE/Ang-II/AT1R. In fact, Ang-1–9 bind AT2R to produce vasodilatory and anti-inflammatory outcomes, and Ang-(1–7) via Mas receptor (MasR) (or Mas-related receptor, Mrg-D) reduces inflammation and blood pressure [71]. This ACE2/Ang-(1–7)/MasR arm also promotes anti-hypertrophic and anti-fibrotic factors by diminishing MAPK- and TGFß-dependent actions, while it stimulates vasoactive prostaglandins and reduces senescence and redox imbalance [72]. Ang-(1–7) enhances NO production by activation of the PI3K/Akt/eNOS pathway and increases PPAR and Src homology region-2 domain-containing phosphatase (SHP)-2 [73]. Parallelly, Ang-(1–7) protects ACE-degradation of bradykinins, which possess potent effects on endothelium-dependent vasodilatation, vascular permeability, and blood pressure control. Interestingly, stimulation of NO by Ang-(1–7) may involve downstream interactions with the bradykinin B2 receptor, AT1R, or AT2R, suggesting a complex interaction of the ACE2/Ang-(1–7)/MasR pathway with other receptor systems [74]. In this line, degradation of Ang-(1–7) is governed predominantly by the amino-terminal catalytic domain of ACE, which catalyzes Ang-(1–7) in Ang-(1–5) [75].

Altogether, both overstimulation and imbalance of the RAAS can be associated with CV responses observed in COVID-19 patients, including hypertension, inflammation atherosclerosis, thrombosis, and myocarditis. Both the CV and systemic ACE/ACE2 ratio could be crucial in the evolution of COVID-19 individuals, favoring either the cardiotoxic ACE/Ang-II/AT1R axis or the cardioprotective ACE2/Ang-(1–7)/MasR pathway [75]. Therefore, an adequate equilibrium of these axes in specific tissues and along the different phases of COVID-19 could ameliorate the severity of disease and mortality in affected subjects. Furthermore, the influence of sex hormones on the RAAS may be of considerable interest for COVID-19 patients. The components of the circulating RAAS as well as those of tissue RAAS are markedly affected by sex hormones, mainly androgens (i.e., testosterone), estrogen (i.e., 17β-estradiol), and progesterone [76]. In general, testosterone seems to increase renin levels and ACE activity, while estrogen elevates angiotensinogen concentrations, decreases renin, ACE, AT1R, and aldosterone, and activates AT2R, MasR, and Ang-(1–7) [76]. Also, progesterone competes with aldosterone for mineralocorticoid receptor. All these actions explain some of the differences between the CV systems of men and women [77]. In this context, epidemiological data showed that men are more prone to die (59–73% of cases) of SARS-CoV-2 infection [78]. A meta-analysis of 61 studies (59,254 patients from 11 different countries) showed an association between the male sex and COVID-19 fatality [79]. In COVID-19, androgens may increase the levels of ACE2 and TMPRSS2 and potentially favor viral uptake into the cells [80]. In contrast, estrogen could trigger different effects on ACE2 levels according to tissue. Estrogen increased ACE2 in cardiac cells [81] but reduced this expression in airway epithelial cells [82]. In any case, ACE2 is located on the X-chromosome and thus, in women, more ACE2 could be free to catalyze Ang-II into Ang-(1–7) after SARS-CoV-2 infection [83]. In addition, androgens generally suppress the inflammatory responses by decreasing the activity of the peripheral blood mononuclear cells, as well as the release of inflammatory factors and cytokines (IL-1β, TNFα, IFN-γ) [84]. In contrast, estrogen can robustly stimulate the innate, humoral, and cellular immunity but in a more balanced and adaptive fashion to the cytokine storm [85]. Therefore, the RAAS can be differentially modulated in males and females affected by COVID-19, leading to distinctive CV consequences. Further research is needed to unveil the potential regulation of the RAAS components by sex hormones on CV cells during COVID-19 development, in men and women, before and after andropause or menopause, respectively.

### 1.3. Influence of ACEi and ARB on SARS-CoV-2 Infection

After SARS-CoV-2 binding to ACE2, the availability of ACE2 to produce Ang-(1–7) could be decreased and the overactive RAAS may shift to the deleterious ACE/Ang-II/AT1R arm [86]. Interestingly, ACE inhibitors (ACEi) are widely prescribed as maintenance therapy in several chronic CV diseases, including arterial hypertension, diabetic cardiomyopathy, and heart failure [87]. Although ACE shares 40–42% identity and 61% homology with ACE2 [88], ACEi can specifically inhibit ACE and the subsequent Ang-II release, likely leading to ACE2 upregulation and Ang-(1–7) formation [89]. Similarly, the angiotensin receptor blockers (ARB) effectively block AT1R, antagonizing the main Ang-II actions and exhibiting protective pleiotropic effects against hypertension and CV inflammation, fibrosis, and thrombosis [87]. Like ACEi, ARB could also rise ACE2 levels, thus augmenting Ang-(1–7) levels. Indeed, ARB attenuated inflammation, oxidative stress, and blood pressure through stimulation of the ACE2/Ang(1–7)/MasR in experimental myocarditis and human hypertension [90,91].

It was originally suggested that elevation of ACE2 might favor SARS-CoV-2 infection and replication in COVID-19 patients with underlying CV disease and ACEi/ARB treatment [92]. However, the European Society of Cardiology and the American Heart Association advise against stopping these maintenance treatments in COVID-19, especially when hypertension or heart failure were present [93]. ACE2 internalization is dependent on heterodimerization with AT1R (under activation with Ang-II), and thus, both ACEi and ARB may prevent ACE2-mediated viral internalization and consequent propagation [94] (Figure 2). In addition, increasing evidence supports advantageous effects of both therapies against CV damage in COVID-19. After more than twenty clinical trials, the available data indicate that ACEi and ARB decrease the viral load, avoid peripheral T cell depletion, and reduce plasma IL-6, CRP, and procalcitonin levels [95,96,97]. ACEi and ARB could diminish the severity of COVID-19, mainly in the hyperinflammatory phase or in subjects with previous CV failures (i.e., hypertension) (Figure 2). Further research is required to analyze whether short- and long-term periods with these therapies may trigger different consequences in specific tissues. In fact, a prolonged use of ARB increased ACE2 expression in kidney, thought this was not maintained in CV cells [98].

### 1.4. ACE2 as a Potential Target for CV Therapy During COVID-19

The full-length ACE2 is highly expressed in the cellular plasma membrane of several tissues, while its extracellular domain generated by proteolytic cleavage can be found in urine and rarely in blood [99]. Interestingly, this soluble isoform can also be a receptor for the S-protein of SARS-CoV-2, acting as a neutralizer [100]. Thus, an increase of plasma ACE2 may reduce SARS-CoV-2 infectivity and COVID-19 evolution (Figure 3). In fact, the administration of recombinant human (rh) ACE2 is being tested against ARDS and acute lung injury [101]. More data about biologic, physiologic, and clinical actions are expected from a new randomized clinical trial, where rhACE2 is intended to block viral entry and replication [102]. In the CV system, by using engineered human blood vessel organoids, rhACE2 was able to reduce the early infection of SARS-CoV-2 [32]. Moreover, increasing plasma ACE2 may also help to balance the RAAS system toward the full-size ACE2 isoform. In fact, infusion of rhACE2 rapidly (48h) triggered high levels of plasma Ang-(1–7) and Ang-(1–5), while it diminished Ang-II and IL-6 concentrations [103,104]. Also, in human explanted hearts from patients with heart failure, administration of rhACE2 normalized Ang-II and augmented Ang-(1–7) and Ang-(1–9) levels (see below) [89]. Ang-(1–9) hydrolyzes more slowly than Ang-(1–7) and could also encourage cardioprotective actions via AT2R [105]. Nevertheless, more studies are required to assess the expression and regulation of ACE2 in other tissues, like liver and intestine, during the different stages of COVID-19. Furthermore, some underlaying comorbidities such as smoking and pulmonary disease can upregulate ACE2 levels in airways, and these levels also increase with advanced age [106]. Besides, type-I and type-III IFN can upregulate ACE2 in human airway epithelial cells, potentially contributing to an increased SARS-CoV-2 binding in upper respiratory tract [107]. However, IFN also promotes tissue-defending properties against viral invasion, and ACE2 could be protective for the CV system by counterbalancing the Ang-II actions [108]. Whether IFN increases soluble ACE2 in circulation is unknown. Nevertheless, beneficial or detrimental effects of IFN may depend on the cell type, stage of infection, presence of co-morbidities, and host sex and age [109]. More preclinical and clinical data are required to clarify IFN’s role in the pathogenesis of COVID-19 and in its potential treatment.

In this line, the ACE2 gene locates in chromosome Xp22 and includes 18 exons with a significant number of polymorphisms. Remarkably, an association of ACE2 single nucleotide polymorphisms (SNPs) with higher susceptibility and prognosis to COVID-19 or to its related CV pathologies has been postulated [110,111]. Four ACE2 variants termed rs4646116, rs148771870, rs762890235, and rs41303171, changed amino-acids 26, 211, 389, and 720 respectively, and presented reduced interaction with the S-protein of SARS-CoV-2 [112]. Similarly, in silico analysis predicted variations in specific amino-acids such as 51 and 355 (N51S and D355N, respectively) that could also induce lower interactions with S-protein [113,114]. In addition, changes in position 351 and 389 (Leu351Val and Pro389His, respectively) could diminish the internalization of the S-ACE2 complex into the host cells [112]. In contrast, alterations of other amino-acids such as 19 and 378 (S19P and H378R, respectively) may trigger higher binding to S-protein [113,115]. Therefore, cellular weakness toward SARS-CoV-2 could be modified with genetic edition of ACE2 sequence [116]. In addition, other SNPs in ACE2 have been associated with common CV pathologies of COVID-19. For instance, rs4646188 and rs879922 correlated with increased LDL-C, cholesterol, and triglycerides [117,118,119] (Table 1), and rs4646188 and rs2106809 variants associated with decreased HDL-C. Development of T2DM was linked to the presence of rs4646188 and rs879922 [120], while rs2106809 and rs2074192 showed higher risk of ventricular hypertrophy in women [121]. Furthermore, the rs2106809 variant was associated with atrial fibrillation in men [122], and rs4646188 and rs4240157 correlated with ischemic stroke [117,118]. Intriguingly, both rs2106809 and rs2074192 were also concomitant with a reduction of circulating Ang-(1–7) [123] (Table 1). Again, these SNPs could be corrected by genetic tools to avoid CV complications in COVID-19 subjects. Additionally, they may be useful to detect more vulnerable COVID-19 patients with higher risk of CV damage.

### 1.5. Ang-(1–7): A Cardioprotective Peptide for COVID-19

Ang-(1–7) can be used as a therapeutic agent of the RAAS that preserves ACE2 expression and activity. Ang-(1–7) can minimize inflammatory tissue damage and disease by reduction of cytokine release and leukocyte recruitment. Thus, we [124] and others [71,125] have recently suggested that activation of MasR can be considered a valid treatment for the deleterious responses prompted by SARS-CoV-2 (Figure 3). Ang-(1–7)/MasR induced cardioprotective actions against hypertension, atherosclerosis, diabetes, heart failure, and stroke by potential downregulation of PI3K/Akt, p38 MAPK, and NF-κB pathways [70]. Since Ang-(1–7) possesses a short half-life in plasma, alternative Ang-(1–7) mimetics (cyclic Ang-(1–7), HPβCD-Ang-(1–7), NPA7, TXA127) and MasR agonists (AVE 0991, CGEN-856, CGEN-857) have been successfully tested in experimental models against oxidation, thrombogenesis, fibrosis, inflammation, endothelial function, and high blood pressure [71,126] (Figure 3). In humans, HPβCD-Ang-(1–7) (hydroxypropyl β-cyclodextrin-Ang-(1–7)) also improved the recovery from a supramaximal physical exercise [127], and NPA7 (a fusion peptide of BNP (B-type natriuretic peptide) and Ang-(1–7)) reduced blood pressure and cardiac uploading [128]. In turn, TXA127 has been catalogued as an orphan drug by the Food and Drug Administration (FDA) for the treatment of pulmonary arterial hypertension, while is being tested in the TXA COVID-19 Clinical Trial to prevent multi-organ failure in patients with moderate to severe COVID-19 [129]. Another trial is evaluating the potential effect of the Ang-(1–7)-derived plasma on mortality of COVID-19 subjects [130], whereas in the Angiotensin-(1,7) Treatment in COVID-19 (ATCO) Trial [131], the efficacy, safety, and clinical impact of intravenous Ang-(1–7) infusion in patients with ARDS will be analyzed. Unfortunately, the CV pathophysiology is not being specifically studied in these trials. We believe that Ang-(1–7) may confer cardioprotective actions in COVID-19 patients with and without underlying CV failures by attenuating the hyperinflammatory and hypercoagulable state.

## 2. Conclusions

CV pathologies are the most prevalent comorbidities during COVID-19 and major causes of morbidity and mortality in patients, even in the absence of ARDS. CV injury may result from lung or systemic pathologies or likely, after direct infection by SARS-CoV-2. Current clinical trials based on antivirals or immunomodulators are aimed to reduce infectivity or related-systemic anomalies. Although their effects on the CV system are not being evaluated, potential deleterious actions might be found (i.e., by antivirals and hydroxychloroquine). While waiting for effective anti-SARS-CoV-2 vaccines, we consider that restoring an adequate balance between the ACE/Ang-II/AT1R and ACE2/Ang-(1–7)/MasR axes of the RAAS could provide positive therapeutic outcomes. ACEi and ARB can be safe and may help to reduce hyperinflammation and viral propagation. Soluble ACE2 might neutralize viral particles in plasma and increase the levels of cardioprotective Ang-(1-9) and Ang-(1–7) peptides. Moreover, gene edition on specific SNPs for ACE2 could reduce viral infectivity and/or related CV complications, preserving the biological activity of enzymes. In this context, local and systemic activation of MasR by specific stimuli (i.e., TXA127, AVE 0991) might counterbalance AT1R-associated actions, such as hypertension, inflammation, and coagulation. Also, sex should be taken into account when designing and analyzing clinical trials of COVID-19, and sex differences may reveal novel therapeutic approaches such as estrogen-related compounds and androgen receptor antagonists. Finally, a potential combination of these strategies could also be tested to better combat both the infective and hyperinflammatory phases of the disease.

## Figures and Tables

**Figure 1 ijms-21-06471-f001:**
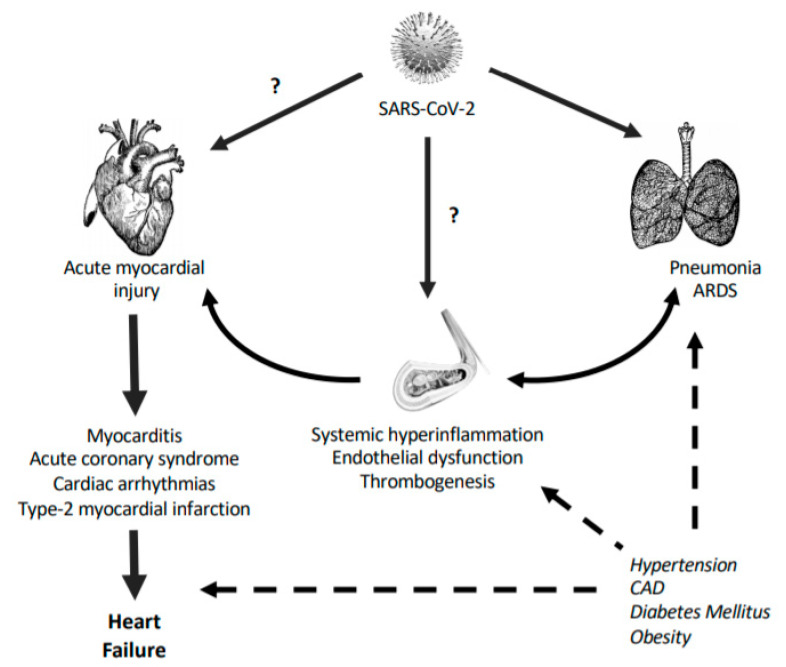
COVID-19 and the cardiovascular (CV) system. The initial step of disease begins with SARS-CoV-2 infection in the upper respiratory system, which leads to fever or cough as the main symptomatology. Then, in some cases, viral infection leads to lung injury characterized by dyspnea, lung inflammation, and pneumonia, with or without associated hypoxia. Later on, if the infection progress, the acute respiratory distress syndrome (ARDS) and extrapulmonary manifestations can appear. Systemic and local hyperinflammation may provoke endothelial dysfunction, vascular permeability, thrombogenesis, and altogether, acute cardiac injury characterized by arrhythmias, acute coronary syndrome, or type-2 myocardial infarction. Also, in COVID-19 subjects, diverse underlaying comorbidities, such as hypertension, coronary artery disease (CAD), and obesity, could accelerate these events. In addition, CV cells may be directly infected by viral particles, reinforcing myocarditis and vasculitis processes. ARDS and CAD stand for acute respiratory distress syndrome and coronary artery disease, respectively.

**Figure 2 ijms-21-06471-f002:**
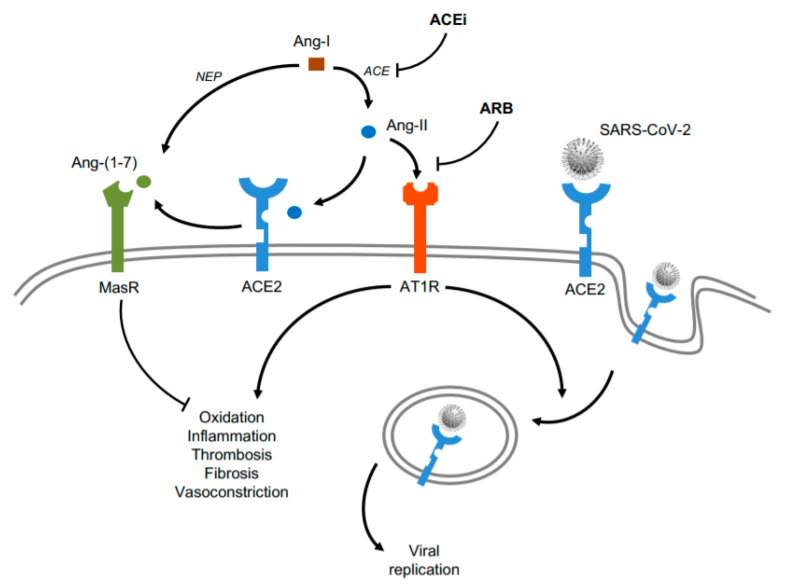
Role of ACEi and ARB in SARS-CoV-2 infection. In COVID-19, stimulation of the ACE/Ang-II/AT1R pathway of the renin-angiotensin-aldosterone system (RAAS) could activate the gene expression of pro-oxidation, pro-inflammation, pro-thrombosis, pro-fibrosis, and vasoconstriction. However, administration of ACEi or ARB drugs may block Ang-II actions, unbalancing the RAAS toward the ACE2/Ang-(1–7)/MasR axis, which conserves cardioprotective properties. Also, in infected cells, ACEi and ARB could reduce AT1R heterodimerization with ACE2, lessening internalization of the ACE2-SARS-CoV-2 complex and viral replication. Ang stands for angiotensin. NEP and ACE mean neprilysin, and angiotensin converting enzyme, respectively. MasR and AT1R stand for Mas receptor and angiotensin receptor type I, respectively. ACEi and ARB mean ACE inhibitors and angiotensin receptor blockers, respectively.

**Figure 3 ijms-21-06471-f003:**
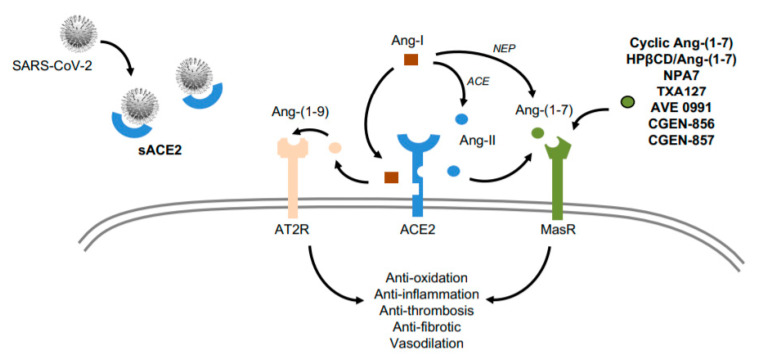
Role of sACE2 and MasR activators in COVID-19. Infusion of the soluble isoform of ACE2 could serve as a decoy receptor for SARS-CoV-2 particles, reducing infectivity while conserving cardioprotective actions of the plasma membrane isoform (via Ang-(1–9) and Ang-(1–7)). On the other hand, some Ang-(1–7) mimetics (cyclic Ang-(1–7), HPβCD-Ang-(1–7), NPA7, TXA127) and MasR agonists (AVE 0991, CGEN-856, CGEN-857) could also trigger protective actions for the CV system in COVID-19 subjects. Ang and ACE stand for angiotensin and angiotensin converting enzyme, respectively. sACE2 and NEP stand for soluble ACE2 and neprilysin, respectively. MasR and AT2R stand for Mas receptor and angiotensin receptor type-II, respectively. Finally, HPβCD-Ang-(1–7) means hydroxypropyl β-cyclodextrin-Ang-(1–7).

**Table 1 ijms-21-06471-t001:** ACE2 variants and related CV alterations. Specific SNPs of ACE2 can associate with abnormalities in lipid profile (total cholesterol (TC), high-density lipoprotein-cholesterol (HDL-C), low-density lipoprotein-cholesterol (LDL-C), or triglycerides (TG)), in blood pressure (hypertension or elevation of systolic blood pressure (SBP)) and CV injury (ischemic stroke, ventricular hypertrophy, or atrial fibrillation), and/or T2DM. Other variants have also correlated with decreased levels of plasma Ang-(1–7) (rs2106809 and rs2074192). All these alterations may worsen COVID-19 evolution. A, G, C, and T stand for adenine, guanine, cytosine, and thymine, respectively.

ID SNPs	Nucleotide Change	Genotype	Related Pathology	Sex	Reference
rs4646188	A > G	rs4646188 (TT + CT)	Increased TC Decreased HDL-C Hypertension	M/F	[117]
		rs4646188 (CC)	Increased TG Increased LDL-C	M/F	[117]
		rs4646188 (TT)	T2DM	M/F	[120]
		rs4646188 (CC + CT)	Ischemic stroke	M/F	[117]
rs2106809	A > G	rs2106809 (TT)	Increased LDL-C Hypertension Reduced Ang-(1–7) Ventricular Hypertrophy	M/F F	[117] [121,123]
		rs2106809 (CC + CT)	Increased TG Decreased HDL-C	M/F	[117]
		rs2106809 (T)	Atrial Fibrillation	M	[122]
rs2074192	C > T	rs2074192 (TT + CT)	Increased TC Hypertension	M/F	[117]
		rs2074192 (TT)	Reduced Ang-(1–7) Ventricular Hypertrophy	F	[121,123]
		rs2074192 (CC)	T2DM	M/F	[120]
rs4240157	C > G, T	rs4240157 (CC + CT)	Increased TC Increased TG Decreased HDL-C Hypertension Ischemic stroke T2DM	M/F	[117,120]
rs1978124	T > A, C, G	rs1978124 (TT + CT)	Increased LDL-C T2DM	M/F	[117,120]
		rs1978124 (G)	Hypertension	M	[119]
rs233575	G > A, C	rs233575 (CC + CT)	Increased LDL-C Increased SBP T2DM	M/F	[117,120]
rs4830542	C > A, G, T	rs4830542 (CC + CT)	Ischemic stroke	M/F	[117]
rs879922	C > G	rs879922 (CC + CG)	Increased LDL-C Increased TC Increased TG Hypertension T2DM	M/F	[117,118,120]
rs4646156	A > C, G, T	rs4646156 (AA + AT)	Increased SBP T2DM	M/F	[120]
rs2048683	T > A, G	rs2048683 (TT + GT)	Increased SBP T2DM	M/F	[120]

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
