# Peer review of "Cardiovascular Damage in COVID-19: Therapeutic Approaches Targeting the Renin-Angiotensin-Aldosterone System"

_ijms, 2020, doi:10.3390/ijms21186471_

Round 1

Reviewer 1 Report

This review by Lumpuy-Castillo et al. addressess the important issue of cardiovascular damage in COVID-19 and explores potential avenues for therapeutic development by targeting the Renin-angiotensin-aldosterone system. In the current scenario this is an important topic to address and will be of considerable interest to the overall scientific community. The following are some areas which the authors have not addressed

  1. The sex-specific effects of AngII and the influence of sex hormones in RAAS system has been of considerable interest in cardiovascular system. For instance it has been shown the metabolites of testosterone and oestrogen have differential effects in the cardiovascular and renal system. Authors have not mentioned how this will affect the RAAS and any potential therapeutic avenues to explore.
  2. AngII and ACE receptor expression is also affected by sex specific hormones which also influences cytokine responses and other physiological effects. Authors could include discussion to address this important area.
  3. Since the SARS-CoV-2 receptor ACE2 has been identified as an interferon-stimulated gene in human airway epithelial cells and is detected in specific cell subsets across tissues, authors could address what effect IFN pathway has on CV and RAAS system and if there are avenues to pursue for therapeutic approach in this regard

Author Response

Responses to reviewer #1:

  1. The sex-specific effects of AngII and the influence of sex hormones in RAAS system has been of considerable interest in cardiovascular system. For instance it has been shown the metabolites of testosterone and oestrogen have differential effects in the cardiovascular and renal system. Authors have not mentioned how this will affect the RAAS and any potential therapeutic avenues to explore.

Thanks for this interesting observation. We agree that sex hormones can regulate the RAAS and thus, may be determinant for both RAAS participation in COVID-19 and for potential new therapies. We have conveniently added a paragraph in page 10 (line 253) with its corresponding references, and have updated the abstract and conclusions of the manuscript.

In general, testosterone seems to increase renin levels and ACE activity, while estrogen elevates angiotensinogen concentrations, decreases renin, ACE, AT1R and aldosterone, and activates AT2R, MasR and Ang-(1-7). Also, progesterone competes with aldosterone for mineralocorticoid receptor. In COVID-19, androgens may increase the levels of ACE2 and TMPRSS2 and potentially favor viral uptake into the cells. In contrast, estrogen could trigger different effects on ACE2 levels according to tissue. In any case, ACE2 is located on the X-chromosome and thus, in women, more ACE2 could be free to catalyze Ang-II into Ang-(1-7) after SARS-CoV-2 infection. Therefore, sex should be taken into account when designing and analyzing clinical trials of COVID-19, and sex differences may reveal novel therapeutic approaches such as estrogen-related compounds and androgen receptor antagonists.

  1. AngII and ACE receptor expression is also affected by sex specific hormones which also influences cytokine responses and other physiological effects. Authors could include discussion to address this important area.

Yes, again, we have included this important issue in the new paragraph in page 10 (line 253). Androgens generally suppress the inflammatory responses by decreasing the activity of the peripheral blood mononuclear cells, as well as the release of inflammatory factors and cytokines. In contrast, estrogen can robustly stimulate the innate, humoral and cellular immunity but in a more balanced and adaptive fashion to the cytokine storm.

  1. Since the SARS-CoV-2 receptor ACE2 has been identified as an interferon-stimulated gene in human airway epithelial cells and is detected in specific cell subsets across tissues, authors could address what effect IFN pathway has on CV and RAAS system and if there are avenues to pursue for therapeutic approach in this regard

We agree with this observation and consequently we have added a new paragraph in page 13 (line 330). IFN can trigger strong anti-viral properties but also, can upregulate ACE2 in airway epithelial cells potentially increasing SARS-CoV-2 infection. However, in the CV system, ACE2 may induce cardioprotective effects by counterbalancing the Ang-II actions. Possibly, a beneficial or detrimental impact of IFN may depend on the cell type, stage of infection, presence of co-morbidities, and host sex and age. Thanks for this interesting point.

Reviewer 2 Report

Cardiovascular damage in COVID-19: therapeutic approaches targeting the reinin-angiotensin-aldosterone system

The present manuscript is a comprehensive review on the mechanisms of SARS-CoV-2 interaction with the cardiovascular system, especially the heart. The authors describe the endothelial damage induced by SARS-CoV-2, as well injury and destruction of heart muscle cells. In this respect the role of the RAAS-system is described in detail, and potential therapeutic targets are outlined and discussed.

The manuscript is well written and comprehensive, and enables the reader to get more insights into the mechanisms with respect to the deleterious effects of SARS-CoV-2 infection, but also with respect to future therapeutic options. There are only some minor questions and remarks:

  • Line 176-178: the authors note that “in this sense there is no evidence of acute coronary syndrome or epicardial plaque rupture, these pathologies can be developed in COVID-19 subject with previous atherosclerosis”. This text passage appears to be somewhat contradictory. It clearly should be stated, whether SARS-CoV-2 is able to induce damage of endothelium and thereby initiate platelet aggregation and consecutive coronary occlusion or not. This of course is of high clinical relevance.
  • “sinus tachycardia” should not be termed as “arrhythmia”, sinus tachycardia rather is a physiological response to various cardiovascular conditions.
  • I recommend to consequently describe all abbreviations outlined in the figures within the figure legend.   

Author Response

Responses to reviewer #2:

  • Line 176-178: the authors note that “in this sense there is no evidence of acute coronary syndrome or epicardial plaque rupture, these pathologies can be developed in COVID-19 subject with previous atherosclerosis”. This text passage appears to be somewhat contradictory. It clearly should be stated, whether SARS-CoV-2 is able to induce damage of endothelium and thereby initiate platelet aggregation and consecutive coronary occlusion or not. This of course is of high clinical relevance.

Yes, as mentioned in the previous section (Endothelial dysfunction), we stated that  COVID-19 can be considered as an eminently endothelial disease though the effects of SARS-CoV-2 on vascular homeostasis are still poorly understood. There are evidence of direct injury on endothelial cells. The endothelium expresses high levels of ACE2, TMPRSS2, and CD147, which may provide a dock-way in cells for viral infection. Post-mortem analysis of COVID-19 subjects revealed the presence of viral structures in vascular beds from different organs. In addition, an increase of endothelial cell death and microvascular platelet-rich thrombotic depositions, together with diffuse endothelial inflammation and accumulation of inflammatory cells have been also detected. In fact, venous thromboembolism in lungs is a common complication in severe COVID-19, and arterial thromboembolism with ischemic events in large vessels or limb have been also reported. However, whether SARS-CoV-2 is a primary cause of plaque destabilization and development of acute coronary syndrome needs to be further investigated.

Thus, we have slightly changed the sentence to: “In this sense, though there is no evidence of acute coronary syndrome or epicardial plaque rupture, these pathologies can be developed in COVID-19 subjects with or without previous atherosclerosis” (page 6, line 173), and we have emphasized this issue with the sentence “Altogether, SARS-CoV-2 may be able to induce damage of endothelium and thereby initiate platelet aggregation and consecutive vessel occlusion” in the previous section (page 6, line 143).

  • “sinus tachycardia” should not be termed as “arrhythmia”, sinus tachycardia rather is a physiological response to various cardiovascular conditions.

Yes, we agree. We have changed the sentence as: “The most common electrocardiographic finding in COVID-19 patients was sinus tachycardia, whereas atrial fibrillation and monomorphic or polymorphic ventricular tachycardia were the most deleterious arrhythmias [62]. Atrial arrhythmias were, however, more frequent among patients requiring mechanical ventilation [63]” (page 8, line 196). Thanks for this clarification.

  • I recommend to consequently describe all abbreviations outlined in the figures within the figure legend.   

Thanks, we have added it in the figures and table